# Spontaneous Vibrations and Stochastic Resonance of Short Oligomeric Springs

**DOI:** 10.3390/nano14010041

**Published:** 2023-12-22

**Authors:** Alexey M. Astakhov, Vladislav S. Petrovskii, Maria A. Frolkina, Anastasia A. Markina, Alexander D. Muratov, Alexander F. Valov, Vladik A. Avetisov

**Affiliations:** 1Semenov Federal Research Center for Chemical Physics, Russian Academy of Sciences, 119991 Moscow, Russia; 2Design Center for Molecular Machines, 119991 Moscow, Russia

**Keywords:** nanosprings, non-linear dynamics, bistability, spontaneous vibrations, stochastic resonance, Duffing oscillators

## Abstract

There is growing interest in molecular structures that exhibit dynamics similar to bistable mechanical systems. These structures have the potential to be used as two-state operating units for various functional purposes. Particularly intriguing are the bistable systems that display spontaneous vibrations and stochastic resonance. Previously, via molecular dynamics simulations, it was discovered that short pyridine–furan springs in water, when subjected to stretching with power loads, exhibit the bistable dynamics of a Duffing oscillator. In this study, we extend these simulations to include short pyridine–pyrrole and pyridine–furan springs in a hydrophobic solvent. Our findings demonstrate that these systems also display the bistable dynamics, accompanied by spontaneous vibrations and stochastic resonance activated by thermal noise.

## 1. Introduction

The increasing focus on nanoscale molecular structures with dynamics reminiscent of bistable mechanical systems is driven by the growing demand for designing and implementing various nanodevices. These devices serve as switches and logic gates [1,2,3,4,5,6], sensors and actuators [7,8,9,10,11,12], and mechanoelectrical transducers and energy harvesters [13,14,15,16,17,18]. Nanoscale bistable systems are equally crucial for validating the principles of stochastic thermodynamics [19,20,21,22]. This field currently aims to expand thermodynamic theories to encompass nanoscale molecular machines [22,23,24,25,26].

This article focuses on nanoscale molecular structures whose dynamic prototypes are the textbook bistable mechanical systems known as Euler arches (a beautiful but simple example of a bistable machine which is composed of two rigid rods joined by a hinge with an elastic spring) [27,28,29] and Duffing oscillators [30,31]. In particular, using molecular dynamics simulations, it was found that the long-term conformational dynamics of short rod-shaped thermosensitive oligomers were similar to the bistable dynamics of a Euler arch [29,32,33], while particular oligomers of a helical form stabilized via weak intermolecular interactions could behave as bistable Duffing oscillators [29,34].

In general, Euler arches and Duffing oscillators can be perceived as mechanical systems whose potential energy is determined via a fourth-degree polynomial function caused by non-linear elasticity of the system elements. The potential energy of these systems has either one minimum or two energy wells separated by a bistability barrier. Therefore, the system can have different dynamic modes controlled by the power loads applied to the system. In particular, the force load compressing a Euler arch controls the arch dynamics, while the stretching of a Duffing spring controls the spring dynamics. Therefore, with driving the power loads, one can operate via the dynamical modes of the system and, accordingly, carry out the transitions between the system discrete states in a controlled manner.

In addition to deterministic transitions, jump-like spontaneous transitions between the two states of a bistable system, known as spontaneous vibrations, can occur due to random disturbances of the system. In the mode of spontaneous vibrations, the time intervals between the jumping (the lifetimes of the system in its states) are random values whose average grows exponentially with the increasing ratio of the bistability barrier to the intensity of the noise, following Kramer’s rate approximation [35]. Therefore, spontaneous vibrations are actually observed when this ratio is not excessively large, such as when the bistability barrier is approximately an order of magnitude higher than the noise intensity. On the other hand, by applying a gentle oscillating force that rocks the bistable potential, spontaneous vibrations can be transformed into almost regular switching between the two states, induced by noise. This phenomenon is referred to as stochastic resonance [36]. Both spontaneous vibrations and stochastic resonance are remarkable manifestations of bistability.

Stochastic resonance is an intriguing phenomenon that arises from the interplay between the bistable dynamics of a system and its stochastic perturbations. Unlike typical noise effects that tend to blur signals, the noise actually amplifies weak signals in the stochastic resonance regime. While the initial concept of stochastic resonance was proposed to explain the regularity of ice ages on Earth [36,37,38], it sparked a proliferation of research exploring its practical applications and interpretations in various macroscopic, global, and even celestial systems [39,40].

In a recent study of a dissipative bistable Duffing oscillator, microscopic phenomena called aperiodic and periodic vacillation were described [41]. These regimes resemble spontaneous vibrations and stochastic resonance, respectively, but arise from different physical nature: spontaneous vibrations and stochastic resonance regimes are well known to be inherent from the thermal noise present in the system, which was not the case explored by [41]. Strictly speaking, spontaneous vibrations occur when forcing amplitude G=0, while stochastic resonance happens when *G* tends to zero and driving frequency Ω corresponds to the mean lifetime in the states (here, *G* and Ω are the designations introduced in [41]).

In recent years, experimental evidence has emerged indicating the potential presence of bistable patterns at sub-micron scales, such as in nanotubes [42,43,44], graphene sheets [15,45], DNA hairpins, and proteins [46,47,48]. It is worth noting that spontaneous vibrations and stochastic resonance in macroscopic mechanical systems, even those as small as a micron, are unlikely to be triggered solely by environmental thermal noise. The bistability barriers in macroscopic systems are much higher than the intensity of thermal noise (∼$k_{B}T$) at natural conditions, necessitating much stronger random perturbations to activate spontaneous vibrations and stochastic resonance, even at the micron scale.

Designing a mechanical system with the non-linear elasticity inherent in a Duffing oscillator is also a non-trivial task. In mechanics, creative combinations of springs have been devised to mimic the Duffing’s bistability (for an example, refer to Lai and Leng [49], Lu et al. [50]). However, in nanoscale mechanics, a potential solution may arise primarily due to strong non-linearity of weak intermolecular interactions whose contribution to the potential energy of a nano-size molecular system may appear significant for collective dynamical modes of the system. An equally important perspective comes from the fact that the bistability barriers of nanoscale systems can be high enough to well separate dynamic states against a background of thermal fluctuations and at the same time, low enough for thermal fluctuations to activate transitions between the well-separated states. A ratio of the bistability barrier to thermal noise intensity (∼$k_{B}T$) of approximately ten to one can serve as a reasonable benchmark. Oligomeric molecules a few nanometers in size appear to be potential representatives of target molecular systems.

Notably, recent intensive molecular dynamic simulations investigating short oligomeric compounds subjected to force loads have uncovered bistable molecules that exhibit the dynamic behavior resembling that of the Euler arches and Duffing oscillators [32,33,34]. These simulations have revealed the presence of mechanic-like bistability in specific oligomeric molecules, accompanied by spontaneous vibrations and stochastic resonance activated by thermal fluctuations. In this paper, we continue our search for bistable nanoscale molecular structures and investigate the behavior of short pyridine–pyrrole (PP) and pyridine–furan (PF) springs in a hydrophobic solvent.

## 2. Materials and Methods

### 2.1. Pyridine–Pyrrole and Pyridine–Furan Springs

PP and PF copolymers (Figure 1a) are conductive polymers consisting of 5- and 6-member heterocyclic rings as synthesized and described by Alan Jones et al. [51] and Alan Jones and Civcir [52], respectively. These copolymers tend to assume a helix-like shape, which is squeezed by the π−π interactions of aromatic groups located at the adjacent turns [53]. Assuming that stacking could lead to non-linear elasticity of the springs and following the quantum calculations of the stacking energy for different configurations of heterocyclic rings [53], the cis-configuration of oligo-PP and oligo-PF with heteroatoms of the 5- and 6-member heterocyclic rings on one side of a polymer chain was selected (see Figure 1a). Guided by the preliminary screening of sizes, we designed two spring models consisting of five monomer units ((oligo-PP-5 and oligo-PF-5) as shown in Figure 1b,c). Both springs were solvated in the hydrophobic solvent, tetrahydrofuran (THF). The distance between the adjacent turns was close to 0.35nm in all non-stretched samples according to Sahu et al. [53].

### 2.2. Simulation Details

Oligo-PP- and oligo-PF-springs and the environmental solvent were modeled in a fully atomistic representation with a canonical (symbol/volume/temperature [NVT]) ensemble (box size: $4.5\times 4.5\times 4.5\;nm^{3}$ for oligo-PP, 7.0×7.0 × $7.0\;nm^{3}$ for oligo-PF) with a time step of 2 fs using Gromacs 2019 [54] and the OPLS-AA [55] force field parameters (for more details, see Parameters for Molecular Dynamics simulation section of Appendix A). The temperature was set at 280 K via the velocity-rescale thermostat [56], which corresponds to the equilibrium state of the springs [53]. Each dynamic trajectory was 300–350 ns long and was repeated three times to obtain better statistics; therefore, the effective length of the trajectories was about one μs for each sample.

The dynamics of the springs were studied by fixing one end of the spring, while the other end was pulled via a force applied along the axis of the spring. The distance (denoted $R_{e}$) between the ends of the spring (yellow and blue balls in Figure 2a) was considered a collective variable describing the long-term dynamics of the spring. Bistability of the spring was specified in agreement with two well-reproduced states of the spring with the end-to-end distances equal to $R_{e}$∼1.10 nm and $R_{e}$∼1.45 nm. These states are referred to as the squeezed and the stress–strain states, respectively.

## 3. Results

### 3.1. Bistable Dynamics of the Oligo-PP-5 Spring

To investigate the dynamics of oligo-PP-5 springs under tension, we initially equilibrated the oligo-PP-5 spring at 280 K with one end fixed. Subsequently, we applied a force $\vec{F}$ along the spring axis to pull the other end. Under weak tensile conditions, the spring’s initial state, compressed by stacking, remained stable, and the spring underwent slight stretching in accordance with linear elasticity. Note that for the oligo-PP-5 spring, we measured not the end-to-end distance, but the distance between the pulled end and the monomer that contacts the pulled end in the squeezed state (see comparison in Appendix A). However, when the pulling force reached a specific critical value of approximately $F_{c}=30\;pN$, the oligo-PP-5 spring exhibited bistability and commenced to vibrate spontaneously. At the critical force value, a junction point emerged, leading to a split into two branches: the branch of zero-stress attractors, representing a stress–strain state, and the branch of unsteady zero-stress states, repelling the dynamic trajectories.

Simultaneously, the squeezed states remain attractive. From the perspective of non-linear dynamical systems, the dynamics of the oligo-PP-5 spring bifurcate at the critical force $F_{c}=30\;pN$. Beyond this critical tensile point, the spring becomes bistable and exhibits spontaneous vibrations, alternating between the squeezed and stress–strain states. The average end-to-end distances of the spring in the squeezed and stress–strain states differ by approximately 0.35 nm, allowing for clear distinction between these two states. Figure 2a shows atom-level snapshots of these two states. Notably, this difference suggests an extension of the stacking pair length to almost twice its original size. As a result, the π−π interactions do not significantly contribute to the elastic energy of the stress–strain states, with the spring’s elasticity mainly determined by the rigidity of the oligomeric backbone.

Figure 2d illustrates the evolution of the statistics of visits to the squeezed and stress–strain states as the pulling force exceeds the critical point $F_{c}$. Below $F_{c}$, the squeezed state represents the sole steady state of the spring. However, at the bifurcation point $F_{c}$, the stress–strain state emerges, rendering the oligo-PP-5 spring bistable, causing it to spontaneously vibrate, although the squeezed state predominates near the critical point $F_{c}$. At F≈75 pN, both the squeezed and stress–strain states are almost equally visited, signifying that the bistability of the oligo-PP-5 spring becomes approximately symmetrical at a considerable distance from the critical point.

In this region, the spontaneous vibrations of the oligo-PP-5 spring are most pronounced. The mean lifetimes of the squeezed and stress–strain states in the spontaneous vibration mode varied in the bistability region, ranging from τ=1–20 ns, depending on the pulling force. In the symmetrical bistability region, neither the squeezed state nor the stress–strain state dominates, resulting in the mean lifetimes of the two states being approximately equal to τ=14 ns (for better understanding of the chemical basis for the observed bistability, see Section “Torsion angle dynamics due to vibrations” of the Appendix A).

Utilizing Kramer’s rate approximation with a collision time for random perturbations ranging from 0.1 to 10 ps, we can roughly estimate the bistability barrier of the oligo-PP-5 spring as ∼10 $k_{B}T$. Noteworthy, the bistability barrier of this bistable oligomeric system is approximately ten times greater than the characteristic scale of thermal fluctuations, $k_{B}T$ [34].

Figure 2c displays a typical trajectory of the long-term dynamics $R_{e}\left(t\right)$ of the oligo-PP-5 spring within the symmetric bistability region. Clear spontaneous vibrations of the spring can be observed without any additional random perturbations applied to activate them. Instead, these vibrations are solely activated via thermal-bath fluctuations. On the other hand, outside the bistability region, non-vibrating trajectories are prevalent.

To investigate the stochastic resonance mode of the oligo-PP-5 spring, we introduced an additional weak oscillating force applied to the pulling end of the spring. This oscillating force was modeled by applying an oscillating electrical field, $E=E_{0}\cos\left(2\pi \nu t\right)$, to a unit charge located on the pulling end of the spring. A counterion was placed at 2.2 nm from the spring center of the mass on the pull axis to balance the system. We need to note that the addition of the charge and a fixed counterion significantly changes the critical force parameters of bistability—for this particular configuration, the system exhibits spontaneous vibrations even without the additional pulling force and reaches symmetrical distribution at F=3.5 pN (for more details, refer to the “Parameters of periodic signal” section in the Appendix A).

Figure 3a presents typical vibrations of the end-to-end distance of the oligo-PP-5 spring in the stochastic resonance mode, along with the power spectrum of these vibrations.

After some preliminary analysis, we decided to examine the frequency response at oscillating field amplitude $E_{0}$ = 0.12 Vnm^−1^ because at amplitudes above $E_{0}>0.15$ Vnm^−1^, the system behavior resembled forced oscillations and not stochastic resonance (the mean lifetime in the state was directly proportional to the external signal period). According to the theory of stochastic resonance [39,40], the primary resonance peak was observed at a frequency of *ν* = ½*τ*, where the period of the applied oscillating field was equal to twice the mean lifetime of the states in the spontaneous vibration mode. We extensively scanned a wide range of oscillating fields to identify the maximal resonance response, as determined by the spectral component at the resonance frequency. Figure 3c,d presents the corresponding results. The maximum resonance response was observed when the period of the oscillating field was close to twice the mean lifetime of the states in the spontaneous vibration mode.

### 3.2. Bistable Dynamics of the Oligo-PF-5 Spring

To investigate the dynamics of oligo-PF-5 springs under tension, we followed the same protocol as for the oligo-PP-5 spring. First, we equilibrated the oligo-PF-5 spring at 280 K with one end fixed. Then, we applied a force $\vec{F}$ along the spring axis to pull the other end. Once again, under weak tensile conditions, the spring’s initial state remained stable and it underwent slight stretching in line with linear elasticity due to the stacking. However, once the pulling force reached a specific critical value, the oligo-PF-5 spring exhibited bistability and began to exhibit spontaneous vibrations in the same way as the oligo-PP-5 spring. As the pulling force reached the critical value, approximately $F_{c}=50\;pN$, a junction point emerged, dividing the system into two branches: one corresponding to the stress–-strain state with zero-stress attractors, and the other representing unsteady zero-stress states that repelled dynamic trajectories. Simultaneously, the squeezed states remain attractive. The average end-to-end distances of the spring in the squeezed and stress–strain states differ by approximately 0.35nm, allowing for clear distinction between these two states. Figure 4a displays atomic level snapshots of these two states. Notably, this difference suggests an extension of the stacking pair length to almost twice its original size. As a result, the π−π interactions do not significantly contribute to the elastic energy of the stress–strain states, with the spring’s elasticity mainly determined by the rigidity of the oligomeric backbone.

Figure 4b,d illustrates the evolution of the statistics of visits to the squeezed and stress–strain states as the pulling force exceeds the critical point $F_{c}$. Below $F_{c}$, the squeezed state represents the sole steady state of the spring. However, at the bifurcation point $F_{c}$, the stress–strain state emerges, rendering the oligo-PF-5 spring bistable, causing it to spontaneously vibrate, although the squeezed state predominates near the critical point $F_{c}$. Within the range of F=125–175 pN, both the squeezed and stress–strain states are almost equally visited, signifying that the bistability of the oligo-PF-5 spring becomes approximately symmetrical at a considerable distance from the critical point.

In this region, the spontaneous vibrations of the oligo-PF-5 spring are most prominent. The mean lifetimes of the squeezed and stress–strain states in the spontaneous vibration mode differed in the bistability region, ranging from τ=1 ns–3 ns, depending on the pulling force. In the symmetrical bistability region, neither the squeezed state nor the stress–strain state dominates, resulting in the mean lifetimes of the two states being approximately equal to τ=2.04 ns. Utilizing Kramer’s rate approximation with a collision time for random perturbations ranging from 0.1 to 10 ps, we can roughly estimate the bistability barrier of the oligo-PF-5 spring as 10 $k_{B}T$. Again, the bistability barrier of this bistable oligomeric system is approximately ten times greater than the characteristic scale of thermal fluctuations, $k_{B}T$.

Figure 4c displays a typical trajectory of the long-term dynamics $R_{e}\left(t\right)$ of the oligo-PF-5 spring within the symmetric bistability region. Clear spontaneous vibrations of the spring can be observed without any additional random perturbations applied to activate them. Instead, these vibrations are solely activated via thermal-bath fluctuations. On the other hand, outside the bistability region, non-vibrating trajectories are prevalent. We have also studied the thermal stability of spontaneous vibrations: in the temperature range of 280–320 K, this regime is stable, preserving the bistability of the spring in the region from F≈50–200 pN while both states are almost equally visited in the range of F=125–175 pN.

To investigate the stochastic resonance mode of the oligo-PF-5 spring, we introduced an additional weak oscillating force by applying an oscillating electrical field, $E=E_{0}\cos\left(2\pi \nu t\right)$, to a unit charge located on the pulling end of the spring. A compensative charge was placed on the fixed end to balance the system (for more details, refer to the “Parameters of periodic signal” section in the Appendix A). Similar to the oligo-PP-5 spring, the adding of a charge and a fixed counterion significantly shifts the bistability region—for this particular configuration, the system exhibits spontaneous vibrations with a symmetrical distribution at F=15 pN. Figure 5a presents typical vibrations of the end-to-end distance of the oligo-PF-5 spring in the stochastic resonance mode, along with the power spectrum of these vibrations.

The primary resonance peak was observed at a frequency of *ν* = ½*τ*, where the period of the applied oscillating field was equal to twice the mean lifetime of the states in the spontaneous vibration mode. We extensively scanned a wide range of oscillating fields to identify the maximal resonance response, as determined by the spectral component at the resonance frequency. Figure 5c,d presents the corresponding results. The maximum resonance response was observed when the period of the oscillating field was close to twice the mean lifetime of the states in the spontaneous vibration mode. In terms of the amplitude of the oscillating field, the maximum resonance occurred at $E_{0}=0.175$ Vnm^−1^. Notably, the resonance response was diminished in the symmetric bistability region at F=15 pN. Beyond this region, the lifetimes of the squeezed and stress–strain states became substantially different, rendering the average lifetime less indicative of the resonance frequency.

## 4. Discussion

The main finding of this work is that spontaneous vibrations and stochastic resonance are not exclusive for oligo-PF springs in water [34], but are present in various compounds and observed in different solvents. However, these bistability effects might vary for different systems; such variations are discussed below.

First note concerns the oligo-PF-5 spring solvated in tetrahydrofuran. In such a system, both spontaneous vibrations and stochastic resonance occur at lesser forces than in water. While in case of water, the spring’s bistability is observed in the region from F≈240–320 pN and the squeezed and stress–strain states were almost equally visited in the region from F=270–290 pN; in THF, the oligo-PF-5 spring exhibits spontaneous vibrations in the region from F≈50–200 pN. Symmetric bistability is achieved at F=150 pN. If we assume that stacking interaction between the turn and the fixed end of the oligo-PF-5 spring governs its elasticity at low tensiles while at high tensiles, the elasticity imposed as the oligomeric backbone becomes dominant, then we must suppose that the π-stacking is destroyed later in water. Such a difference can be explained by the fact that oligo-PF springs are hydrophobic and prefer to remain in the squeezed state in a hydrophilic environment, i.e., water.

The second note concerns the overstretching limit of the oligo-PF-5 spring. Our previous findings show that the overstretching of oligo-PF-5 springs in THF or vacuum occurs at F=275 pN, while in water, it happens at F=330 pN (see [34], Appendix A). Moreover, at F=330 pN, all three states, the squeezed, the stress–strain, and the overstretched, coexist, so one might assume that the bistability region extends up to the overstretching limit. However, our results of modeling the oligo-PF-5 spring in THF do not support this idea: the bistability ceases at F=200 pN and at larger forces, the spring exists at the stress–strain state up to the overstretching limit.

The next note concerns the bistability of the oligo-PP-5 spring. For this system, we observe spontaneous vibrations in the region from F≈30 to 80 pN and the symmetric bistability is achieved at F=75 pN. Again, above the force of F=80 pN, the oligo-PP-5 spring exists in a stress–strain state up to the overstretching limit, which was F=305 pN in this case. Interestingly enough, we could not detect the bistable behavior of the oligo-PP-5 spring in water (See “Oligo-PP-5 in water” section of the Appendix A). This might be due to the fact that although the bistable behavior of the oligo-PP-5 spring might be theoretically expected, the overstretching occurred earlier than the transition from its squeezed state to the stress–strain state since these two phenomena are independent from each other. These three observations once again support the idea that the interplay between short-ranged (π-stacking, for instance) and long-ranged couplings is crucial for the observation of the bistable behavior. Fine tuning of the setup might be required whether one desires to reproduce these results experimentally.

The last note concerns the stochastic resonance of both oligo-PF-5 and oligo-PP-5 springs. If an external oscillating force is strong enough, the forced oscillations of the spring might also happen. These forced oscillations have different origins than the stochastic resonance, the frequency of which is governed by the lifetime of the states in the spontaneous vibrations regime. It is important to distinguish between forced oscillations occurring at large amplitudes of the oscillating field and stochastic resonance observed at weak amplitudes. Based on our results, we determine $E_{0}=0.175$ Vnm^−1^ as the limit of the amplitude; below this limit, the stochastic resonance was established.

## 5. Conclusions

We conducted atomic level simulations on short PP- and PF-springs under stretching conditions and the results revealed clear indications of bistable dynamics characteristic of Duffing oscillators. During the study, we fixed one end of the springs while pulling the other end along the spring’s axis. We observed typical bistability characteristics, such as spontaneous vibrations and stochastic resonance, in both springs. To explore the symmetrical bistability conditions thoroughly, we examined a wide range of controlling parameters and determined the mean lifetime of the states in the spontaneous vibration mode for each spring.

By using Kramer’s rate approximation with collision times ranging from 0.1 to 20 ps, we estimated the bistability barriers for both springs to be within the range of 5 to 15 $k_{B}T$. Remarkably, the time scales of spontaneous vibrations and the bistability barriers for oligo-PP-5 and oligo-PF-5 springs were found to be similar to those of the oligomeric Duffing oscillator and oligomeric Euler arch, as described in previous studies [32,33,34]. The high bistability barriers of these short oligomeric springs effectively prevent separation of the two states due to thermal noise. However, at the same time, these barriers allow transitions between the states to be activated via thermally enriched fluctuations with higher energy.

Based on our modeling of short PP- and PF-springs, along with the previous modeling of oligomeric Euler arches, we propose that nano-sized oligomeric structures stabilized by short-range, low-energy couplings (e.g., weak hydrogen bonds, hydrophilic-hydrophobic interactions, and π−π interactions) can indeed exhibit bistability, accompanied by thermally-activated spontaneous vibrations and stochastic resonance. However, an experimental validation of our findings is a big challenge and we might only speculate on the possible use of atomic force microscopy [57] or optical tweezer techniques [58], which in general may probe the necessary forces and distances but lack time resolution.

## Figures and Tables

**Figure 1 nanomaterials-14-00041-f001:**
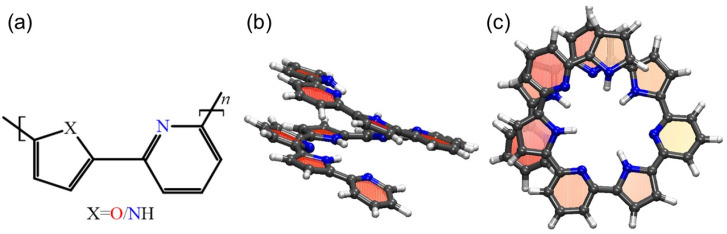
Pyridine–furan (PF) and pyridine–pyrrole (PP) springs with five monomer units (oligo-PF-5 spring and oligo-PP-5 spring, respectively): (**a**) Chemical structure of a pyridine–furan (*X* represents oxygen, *O*)/pyridine–pyrrole monomer (*X* represents NH group) unit with heterocyclic rings in cis-configuration. (**b**) Front and (**c**) top views of an oligo-PP-5 spring in the atomistic representation. The spring has one complete turn consisting of approximately 3.5 monomer units.

**Figure 2 nanomaterials-14-00041-f002:**
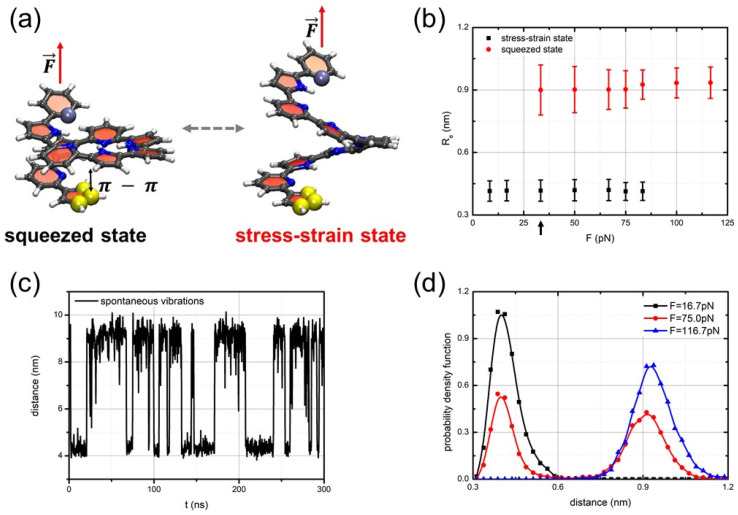
(**a**) Computational models of the oligo-PP-5 system with applied longitudinal load. The squeezed and the stress–strain states of the spring are shown on the right and left, respectively. The pulling force, *F*, is applied to the top end of the spring; (**b**) the state diagram shows a linear elasticity of the oligo-PF-5 spring up to $F_{c}\approx 30\;pN$ (see the black arrow) and bistability of the spring in the region from F≈30–80 pN; (**c**) spontaneous vibrations of the oligo-PP-5 spring at F≈75 pN; and (**d**) evolution of the probability density for the squeezed and stress–strain states when pulling force surpasses the critical value.

**Figure 3 nanomaterials-14-00041-f003:**
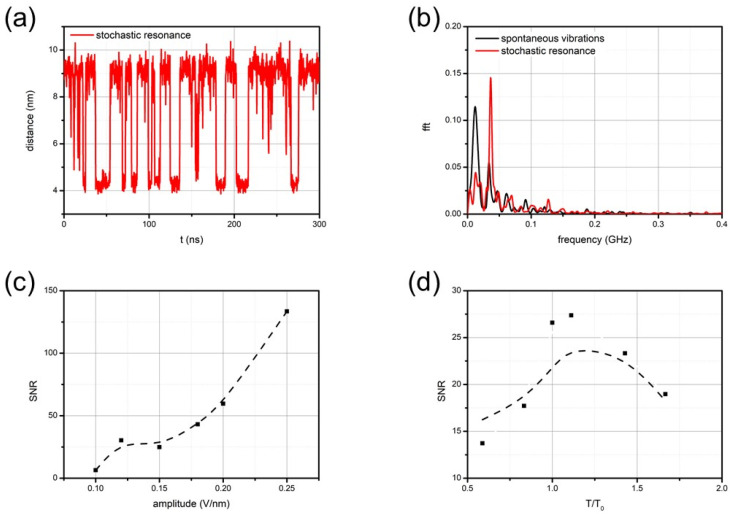
Stochastic resonance of the oligo-PP-5 spring induced by an oscillating field $E=E_{0}\cos\left(2\pi \nu t\right)=E_{0}\cos\left(2\pi t/T\right)$: (**a**) the dynamic trajectory at F=3.5 pN, T=28 ns, and $E_{0}=0.12$ Vnm^−1^; (**b**) power spectrum of spontaneous vibrations (black curve) and stochastic resonance (red curve); (**c**) the dependence of the main resonance peak amplitude on $E_{0}$ (T=28 ns); and (**d**) the dependence of the main resonance peak amplitude on the period *T* of oscillating field ($E_{0}=0.12$ Vnm^−1^).

**Figure 4 nanomaterials-14-00041-f004:**
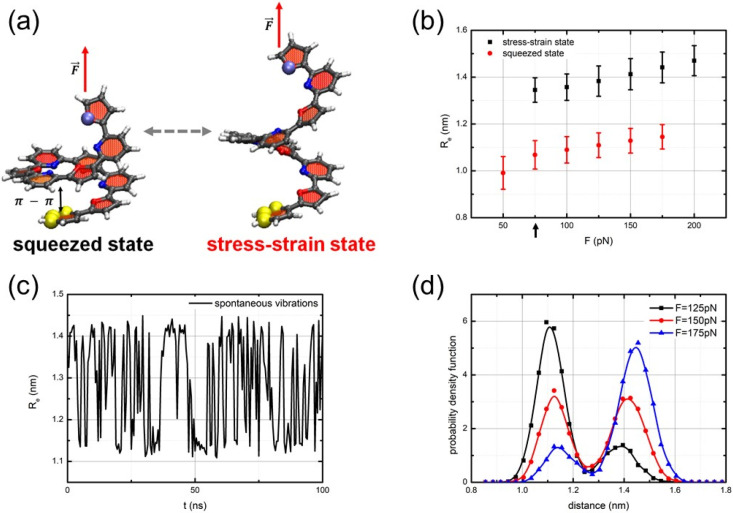
(**a**) Computational models of the oligo-PF-5 system with applied longitudinal load. The squeezed and the stress–strain states of the spring are shown on the left and right, respectively. The yellow spheres at the lower end of the spring indicate the fixation of the pyridine ring via rigid harmonic force. The pulling force, *F*, is applied to the top end of the spring. (**b**) The state diagram shows a linear elasticity of the oligo-PF-5 spring up to $F_{c}\approx 50\;pN$ (see the black arrow) and bistability of the spring in the region from F≈50–200 pN; (**c**) spontaneous vibrations of the oligo-PF-5 spring at F≈150 pN; and (**d**) evolution of the probability density for the squeezed and stress–strain states when pulling force surpasses the critical value.

**Figure 5 nanomaterials-14-00041-f005:**
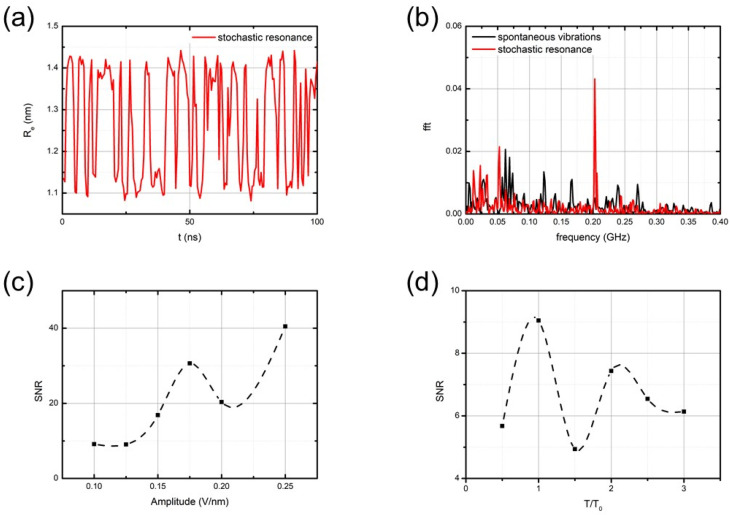
Stochastic resonance of the oligo-PF-5 spring induced by an oscillating field $E=E_{0}\cos\left(2\pi \nu t\right)=E_{0}\cos\left(2\pi t/T\right)$: (**a**) the dynamic trajectory at F=15 pN, T=4.9 ns, and $E_{0}=0.175$ Vnm^−1^; (**b**) power spectrum of spontaneous vibrations (red curve) and stochastic resonance (black curve); (**c**) the dependence of the main resonance peak amplitude on $E_{0}$ ($T_{0}=4.9\;ns$); and (**d**) the dependence of the main resonance peak amplitude on the period *T* of oscillating field ($E_{0}=0.175$ Vnm^−1^).

## Data Availability

The data presented in this study are available on request from the corresponding authors.

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
