# Peer review of "Spontaneous Vibrations and Stochastic Resonance of Short Oligomeric Springs"

_nanomaterials, 2023, doi:10.3390/nano14010041_

Round 1

Reviewer 1 Report

Comments and Suggestions for Authors

Please see the attached PDF

Comments on the Quality of English Language

I recommend minor editing

Author Response

We greatly appreciate the feedback provided by the reviewer.

Regarding the suggestions to enhance the rigor and clarity of the study:

  • Up to the moment, we have conducted more than one thousand simulation runs during which we observed spontaneous vibrations, not only for the present study, but also for our previous works (Avetisov et al., Nanomaterials 2021) and future research. During our investigation, we haven’t encountered any specific limitations than those imposed by the all-atom molecular dynamics (MD). As for an outline of possible experimental validation, we might only speculate on the possible use of atomic force microscopy or optical tweezer techniques. We have included the corresponding passage in the conclusion section (lines 332-335)
  • We agree that the influence of hydrophobic solvents on the bistability is of particular interest and further exploration would be desirable. At the same time, we believe that such a detailed investigation involves using ab initio simulations along with molecular dynamics, which is far out of the scope of the current paper. The main findings of it are that the change of water to tetrahydrofuran has preserved the effects of spontaneous vibrations / stochastic resonance but affected the value of critical force Fc.
  • Using Kramer’s rate approximation is an ordinary method when describing spontaneous vibrations and stochastic resonance (in the text we refer Gammaitoni et al., Rev. Mod. Phys. 1998, Wellens et al., Rep. Prog. Phys. 2004) to estimate bistability barrier.
  • We discuss the potential applications of the molecular structures in two-state systems in gereral way. No doubt that there exist specific arguments that such systems might find many applications, but such discussion might be more appropriate in a more general review.
  • A comparison of bistability at the nanoscale with that seen in macroscopic systems is an interesting issue but is out of the scope of the current work. Here we only state that nanoscopic systems can exhibit bistability and give two examples of such systems (in addition to our previous paper Avetisov et al., Nanomaterials 2021). This may be of interest for scientific society since in our example the bistable effects of spontaneous vibrations / stochastic resonance are activated by a conventional thermal noise.
  • In the current study we are limited to the restrictions of the force-field we use. It is well-established in a temperature range around room temperature (300K), and at the same time, tetrahydrofuran’s boiling point is 330K. However, our investigation has shown that within this narrow temperature range 280K-320K the nanospring’s behavior changes a little (if any). We have added the corresponding statement in the text (lines 239-242)
  • Our work may give rise to the following directions of investigation: further studies of different solvents; additional research on how the amplitude and frequency of a weak oscillating force influence the stochastic resonance; how nanosprings may synchronize their behavior when bound. Such an outlook is our sketch view and we decide to omit it from the text.

Regarding the technical suggestions:

  • Thank you very much for the suggestion to compare spontaneous vibrations with aperiodic/periodic vacillation. We have added a paragraph on such comparison; unlike these phenomena, spontaneous vibrations have stochastic nature, arise from the thermal noise present in the system and occur when forcing amplitude G=0 (lines 57-65).
  • Accordingly, since the nature of spontaneous vibrations is stochastic, stochastic resonance occurs when G->0 and driving frequency accepts a specific value corresponding to the mean lifetime in the states (lines 57-65).
  • We have added clarification regarding the statement about the ratio of the bistability barrier to thermal noise intensity (line 84).
  • Thank you very much for noting that there are no figure labels. Indeed, early versions of figures were attached at the time of submission. We apologize for that; we have addressed the issue and uploaded the correct version with correct labelling.
  • We have modified the figures indicating the critical force.

We appreciate the reviewer's valuable feedback.

Reviewer 2 Report

Comments and Suggestions for Authors

The authors present a beautiful molecular dynamics study on the behaviour of molecular springs. The paper probably deserves publication in Nanomaterials, however, two points have to be amended:

- I would request that the authors study in more detail the chemical basis for the observed bistability. I expect to see the change of the most relevant angles or dihedral angles with time.

- The Supplementary Material does not seem to fit to the main part of the manuscript. Was the wrong file uploaded?

Author Response

We greatly appreciate the feedback provided by the reviewer.

  • We have calculated the correlation between the most relevant dihedral angles and the distance between stacking pairs (lines 164-166) and added the corresponding Figure to the SI section.
  • Thank you very much for noting that the SI does not correspond to the main text. Indeed, an early version of SI was attached at the time of submission. We apologize for that; we have addressed the issue and uploaded the correct version.

We appreciate the reviewer's valuable feedback.

Reviewer 3 Report

Comments and Suggestions for Authors

In this manuscript the authors present a theoretical study on the use of bistable systems, the isomerizing large organic molecules, as molecular switches. 

They present fairly good studies of the quantum chemistry part associated to the Born-Oppenheimer states of the molecular systems.

Later the simulate the dynamic between the two isomer wells existing in the two molecular systems.

As such, the study is original and interesting, and certainly deserves publication.

A first question comes here to my mind: Since the systems under study are molecules, rather that Nanodevices or nanostructures, should not the paper get more visibility in the MDPI journal Molecules than in here? I do not have a cear opinion in this matter, so a let the final decission up to the authors and/or editor.

The paper is well explained and written, and can be followed quite easily. Especially good is the Introduction.

However, I have a number of points, that when clarified, should contribute to improve the manuscript:

1) In the second paragraph of the Intro the authors refer to Euler arches (EAs), and give references to two catastrophe books. The referee has not finf any reference to EAs neither in the literature nor in the two mentioned books. Accordingly, I recommend to the authors to fix this, and/or add a short paragrapgh describing EAs and their relation with the problem they treat.

2) The literature is relatively old, only a few references are 2020 and two more from 2021. Could the authors fix this?

3) I would like to have found more details of the calculations in sect 2.2.

4) also in this sect. why the molecule end is kept fixed? what are the authors simulating here?

5) In line 155 the authors mention in passing Kramer's theory. I guess that a more thorugh discussion should be placed here, especially regarding the present problem.

Also what is friction and stochastic force in their problem?

6) In paragrapph starting at line 164 theauhos introduce an additional weak force. How reallistic is that? and again, what are they simulating?

7) atthe end of the second paragraph in Discussion an hydrophobic environment is mentionad. It is not clear to this referee what the authors meant by that. Could they explain that a little better?

Summarizing, in my opinion the manuscrit can be published after the above point have been taken care of.

Comments on the Quality of English Language

The quality of the English is fair.

Author Response

We greatly appreciate the feedback provided by the reviewer.

0) Regarding the choice of journal:

We appreciate the reviewer's concern about the choice of journal for our work. We carefully considered the selection of Nanomaterials based on its focus on nanoscale materials and their applications. The nanosprings in our study are indeed nanoscale materials, and we believe that their behavior and properties are of interest to the Nanomaterials community. The subject

  • We have added a short description of EA along with additional references to our previous works where EAs were addressed in much more detail (lines 21-23).
  • We have updated the literature including more recent publications.
  • More details on the calculations are present in the SI section.
  • The bottom end of the nanospring is fixed so that it would not move because of the action of pulling force. The molecule must be under an external load to exhibit spontaneous vibrations.
  • Using Kramer’s rate approximation is an ordinary method when describing spontaneous vibrations and stochastic resonance and we follow the established tradition. We do not account for complex effects, and for our tasks it is sufficient to use Kramers rate to estimate the bistability barrier. More detailed description can be found in our previous paper (Avetisov et al., Nanomaterials 2021) which we refer in the text; friction and stochastic forces are also concerned there. We have added a corresponding reference to the text.
  • An addition of weak oscillating force is necessary for the observation of stochastic resonance by its definition.
  • Regarding the hydrophilic environment we referred to our previous study that concerned the behavior of a nanospring in water (line 280).

We appreciate the reviewer's valuable feedback.

Round 2

Reviewer 2 Report

Comments and Suggestions for Authors

The authors performed the required changes. They found a correlation between the total distance and a single dihedral angle. While they could have investigated the chemical foundations of the observed bistability more deeply, I believe, the manuscript is acceptable now. In future work one would like to see a confirmation using higher-level methods such as ab initio molecular dynamics.

Minor point: Supplementary Material: (Figure S3a, blue curve) should read (Figure S2a, blue curve)